# High Prevalence of *Rickettsia raoultii* Found in *Dermacentor* Ticks Collected in Barnaul, Altai Krai, Western Siberia

**DOI:** 10.3390/pathogens12070914

**Published:** 2023-07-06

**Authors:** Alexey V. Rakov, Tatiana A. Chekanova, Ketevan Petremgvdlishvili, Andrey V. Timonin, Anna V. Valdokhina, Sergey V. Shirokostup, Natalia V. Lukyanenko, Vasiliy G. Akimkin

**Affiliations:** 1Laboratory for Natural Focal Infections Epidemiology, Central Research Institute of Epidemiology, 111123 Moscow, Russia; tchekanova74@mail.ru (T.A.C.); ketevan0511@mail.ru (K.P.); 2Department of Epidemiology, Microbiology and Immunology, Altai State Medical University, 656038 Barnaul, Russia; sekttor@mail.ru (A.V.T.); natavalluk@mail.ru (N.V.L.); 3Scientific Group of Biotechnology and Genetic Engineering, Central Research Institute of Epidemiology, 111123 Moscow, Russia; valdokhina@cmd.su; 4Institute of Public Health and Preventive Medicine, Altai State Medical University, 656038 Barnaul, Russia; shirokostup@yandex.ru; 5Central Research Institute of Epidemiology, 111123 Moscow, Russia; vgakimkin@yandex.ru

**Keywords:** ticks, spotted fever group *Rickettsia*, phylogenetic analysis, Russia

## Abstract

The prevalence of the tick-borne spotted fever group rickettsioses pathogens in ticks collected in Barnaul, the administrative center of Altai Krai, Western Siberia, was studied. The causative agent of tick-borne lymphadenopathy (TIBOLA) *Rickettsia raoultii* was revealed to be present in 61.9% of the samples from *Dermacentor* ticks. Moreover, *Rickettsia helvetica* has been identified in 5.1% of *Ixodes* ticks.

## 1. Introduction

Altai Krai is endemic with tick-borne rickettsioses (TBRs) in the Russian Federation. This Asian region, as well as the adjacent areas of Russia (Novosibirsk and Kemerovo Regions, Altai Republic), and neighboring Kazakhstan are hotspots for tick-borne infections, particularly for TBR [1]. Several human pathogenic genospecies of the tick-borne spotted fever group rickettsioses (SFGRs) can be causative agents of TBR. The endemicity of this region is due to the abundance and diversity of the local ixodid tick fauna. Epidemiological surveillance and control of TBR is a difficult task. In general, the disease is diagnosed in humans by considering a combination of clinical symptoms, the most important of which are a fever and a rash, as well as taking into account epidemiological data (seasonality and/or tick infestation/attachment). It should be noted that a rash is not always observed in TBR, which is a feature of rickettsiosis commonly found in Siberia, known as North Asian tick typhus (NATT, synonym: Siberian tick typhus), caused by *Rickettsia sibirica* subsp. *sibirica* (*R. sibirica* sensu stricto). Until recently, NATT was considered the disease with the highest incidence among rickettsioses in Russia [2].

The role of *R. raoultii* (synonym: *R. conorii* subsp. *raoultii*, https://lpsn.dsmz.de/subspecies/rickettsia-conorii-raoultii) in the etiology of the infection known as tick-borne lymphadenopathy (TIBOLA/DEBONEL/SENLAT) in patients affected by tick bites has been demonstrated recently in the Novosibirsk region. Clinically, the disease presents with an intermittent fever of up to 37.7 °C and the absence of a rash [3]. It should be emphasized that NATT and tick-borne lymphadenopathy have different clinical manifestations. In the latter case, rickettsiosis often occurs without a rash. Patients without a rash who have been bitten by a tick are often not noticed by physicians. In addition, the same patients may have been infected with other SFGR genospecies/genovariants [4]. The difficulty in determining the etiologic agent from clinical material is also due to the fact that serologic tests are group-specific and molecular biology tests are time-limited in patients who respond to antibiotic use. There is a clear need to study the diversity of *Rickettsia* species in vectors in areas endemic with tick-borne infections.

The aim of this study was to determine the prevalence of SFGR in ticks from natural biotopes of Barnaul, the administrative center of Altai Krai, with a population of approximately 630,000 people (30% of the region’s population).

## 2. Materials and Methods

### 2.1. Study Area and Ticks

Barnaul (geographic coordinates of center: 53°20′ N 83°45′ E) is the largest city and administrative center of Altai Krai, the Asian part of Russia, located in the forest steppe zone of the West Siberian Plain on the left bank of the Ob River. The border with Kazakhstan is 345 km (210 mi) to the south, which makes Barnaul the closest major city to the Altai Mountains. The city is relatively close to the Russian borders with Mongolia and China (Figure 1).

The climate of Barnaul is continental with a short, warm, and humid summer. The relative humidity during the warm period is around 62%. The native vegetation of Barnaul and its suburbs belongs to the southern forest steppe zone and is represented by steppe, forest, and floodplain-meadow types. Grasses and motley grasses are common here: narrow-leaved bluegrass, Russian brome grass, silver cinquefoil, sickle alfalfa, etc.

All 300 ticks of the three genera of the *Ixodidae* family were collected in the urban parks, squares, and wastelands within the city of Barnaul, Altai Krai. In total, there were seven zones for tick collection: urban ecosystem with grass plantations (1, 2), urban ecosystem near buildings next to coniferous forest (3), urban forest park area (4), boundary between aquatic and forest ecosystems (5), and boundary between urban and forest ecosystems (6, 7). Figure 2 depicts the collection zones and geographic coordinates. Ticks were collected by flagging the vegetation between April and June 2022. Briefly, ticks were collected in the daylight hours by dragging a flag 1.5 × 2.0 m in size over vegetation in above-mentioned zones including both coniferous and broad-leaved trees with a moderate herb layer. All ticks were unengorged (unfed). Ticks attached to the flag were removed, placed into individual Eppendorf tubes, and stored at −70 °C until transportation to the laboratory. Transportation to the laboratory was carried out by air in a thermal container with enclosed ice packs for 3 days. Homogenization and DNA extraction were carried out within a week from the day that they arrived at the laboratory. The isolated DNA and the remains of the homogenates were stored at −20 °C during the month when PCR and sequencing were performed. Subsequently, all nucleic acid residues and homogenates were transferred for long-term storage at −70 °C.

### 2.2. DNA Extraction and Quantitative PCR

After morphological identification, each tick was individually washed with 96% ethanol and then 0.15 M NaCl solution. Ticks were homogenized in a 2.0 mL Eppendorf tube in 300 μL 0.15 M NaCl solution with tungsten carbide beads in a TissueLyser LT homogenizer (Qiagen, Hilden, Germany) at 50 Hz/s for 10 min. Total DNA was extracted using «AmpliSens^®^ RIBO-prep» kit (CRIE, Moscow, Russia). qPCR screening for *Rickettsia* spp. was performed using Rotor-Gene Q (Qiagen, Hilden, Germany) and «AmpliSens^®^ *Rickettsia* spp. The SFG-FL» kit targeted the *ompB* gene (CRIE, Moscow, Russia) according to the manufacturer’s instructions.

### 2.3. Sequence Analysis

The SFGR species were determined by Sanger sequencing amplified by conventional PCR to identify citrate synthase *gltA*, outer membrane protein A *ompA*, outer membrane protein B *ompB*, 17 kDa protein *htrA*, and 16S rRNA gene fragments on two DNA strands using primers listed in Table 1. BLASTN 2.13.0 was used to search against the GenBank non-redundant database using default parameters. Dendrograms were constructed in MEGA 6.06 using the maximum likelihood method on aligned gene fragment sequences with a bootstrap value of 1000. Homologous DNA sequences from the complete genomes of the corresponding representative SFGR obtained from GenBank were used for comparison and the *R. bellii* An04 genome (NZ_CP015010) was used as an outgroup where possible.

Confidence intervals were calculated by using the modified Wald method in QuickCalcs (GraphPad, San Diego, CA, USA).

The sequences from this study are available in GenBank (OQ723808-OQ723947, OQ727556-OQ727559).

## 3. Results

Ticks from three genera (*n* = 300) were collected in this study. After morphological identification at the genus level, the majority of ticks were found to belong to the genus *Dermacentor* (*n* = 202), mostly *D. nuttalli* and *D. silvarum*. There were significantly fewer ticks belonging to the genus *Ixodes* (*n* = 97) and one tick belonging to the genus *Haemaphysalis*.

For primary screening, we used SFGR-specific qPCR. Positive qPCR results were obtained in 43.3% of samples (130/300). A total of 96% (125/130) of all positive qPCR results were obtained from *Dermacentor* ticks. A total of 115 of 125 rickettsial DNA samples with Ct (cycle threshold) values ≤ 31 isolated from ticks were subsequently determined by Sanger sequencing.

To identify *Rickettsia* species, each sample was additionally tested by conventional PCR with electrophoretic detection, and fragments of the *gltA*, *ompA*, *ompB*, *htrA*, and 16S rRNA genes of *Rickettsia* spp. SFG were sequenced. GenBank analysis of rickettsial DNA sequences found in 115 *Dermacentor* ticks confirmed the species of the ticks themselves with 100% identity and the *gltA* gene fragment of 69 strains of *R. raoultii*, including three complete genomes of Khabarovsk, IM16, and BIME strains (Figure 3A). There was also confirmation with 100% identity of the *ompA* gene fragments of the *R. raoultii* isolates from *D. marginatus* in Turkey and Italy (Figure 3B). Table 2 shows the distribution of SFGR-positive samples among the three tick genera. For all five genes, the isolates obtained were part of the *R. raoultii* cluster but differed slightly from the three complete genome strains of the *ompB* gene fragment (Figure 3 and Figure 4A–C). Therefore, we concluded that 100% of *Rickettsia* spp. SFG identified in *Dermacentor* ticks belonged to the same species, *R. raoultii*. Furthermore, the sequences of the fragments of all the five genes mentioned above were 100% identical.

Notably, DNA samples of *Rickettsia* isolated from three *Ixodes* spp. ticks, amplified and sequenced by fragments of the *gltA* gene, showed the highest identity with the homologous gene of strain *R. helvetica* C9P9 (99.74%) (Figure 3A) and the fragments of *ompB*, *htrA,* and 16S rRNA (Figure 4A–C). As shown in Figure 4A–C, *R. helvetica* was 100% identical to *R. helvetica* C9P9 for *htrA* and 16S RNA gene fragments and 99.77% homologous to the *ompB* gene. Similar to some SFGR species, *R. helvetica* also lacks the *ompA* gene.

## 4. Discussion

In Russia, the incidence of the population with NATT and Astrakhan spotted fever, the etiological sources of which are *R. sibirica* subsp. *sibirica* and *R. conorii* subsp. *caspia*, respectively, is subject to official recording. The regions of Siberia are endemic with NATT, which occurs with a rash and a fever. However, the absence of a rash among tick-bite patients in the region does not rule out *Rickettsia* infection. According to official data from Rospotrebnadzor, in 2021, 9952 people suffered from tick bites in the Altai Krai (436.06 per 100,000 population) and NATT was diagnosed for 460 people (including 299 people from rural areas) [9]. There is no registration of rickettsioses without a rash. However, such rickettsiosis cannot be excluded.

Barnaul’s recreational zones are located on the territory of the Ob Plateau, which is mostly covered with grassy meadows and feather-grass steppes. As a rule, *Dermacentor* ticks live in such areas. Most often, the hosts for these ticks are small mammals.

In this study, the SFGR pathogens were found in ticks in Barnaul at a high prevalence. Ticks of the genus *Dermacentor* are the main host and natural reservoir of *R. raoultii* not only in Europe [10] but also in some Asian countries [11]. All rickettsial DNA isolated from *D. reticularis* in Europe was found to belong to *R. raoultii*, with an average infection rate of 47.9% [12]. In a previous study, the percentage of *R. raoultii* in questing *Dermacentor* ticks in Altai Krai was relatively low (15.0%) [13]. In our study, all *R. raoultii* isolates were detected in the vast majority (61.9%, 95% CI 55.0%–68.3%) of the *Dermacentor* spp. ticks. Considering that this *Rickettsia* species is thought to be the cause of DEBONEL/TIBOLA/SENLAT syndrome and does not cause the classic rash clinic typical for most TBRs registered in Russia (NATT, Far Eastern tick-borne rickettsiosis, Astrakhan spotted fever), further monitoring studies are necessary.

*R. helvetica* has also been identified in 5.1% (95% CI 1.9–11.8%) of *Ixodes* ticks. It was previously recognized as non-pathogenic in humans, but several cases described in Sweden suggest that infection may be accompanied by non-specific human fever, meningitis, and perimyocarditis [14,15,16]. Speck et al. [17] classified *R. helvetica* as a low-pathogenic microorganism. We did not find the NATT causative agent, *R. sibirica*, in tick vectors in this study. Although Altai Krai is endemic with TBR, it has been understudied in terms of understanding the genetic diversity of human pathogens in ixodid ticks. Information about the prevalence of *Rickettsia* species/subspecies in this particular area should be taken into account by clinicians since rickettsioses can occur without a rash.

## 5. Conclusions

Analysis of the amplified sequences of the *gltA*, *ompA*, *ompB*, *htrA*, and 16S rRNA regions of *Rickettsia* spp. suggests that the detected *R. raoultii* and *R. helvetica* represent homogeneous populations but are genetically distinct from previously described genotypes isolated from ticks in other regions of Russia. It is necessary to continue the study of the species diversity of pathogens contained in ticks from the Russian regions endemic with tick-borne infections.

## Figures and Tables

**Figure 1 pathogens-12-00914-f001:**
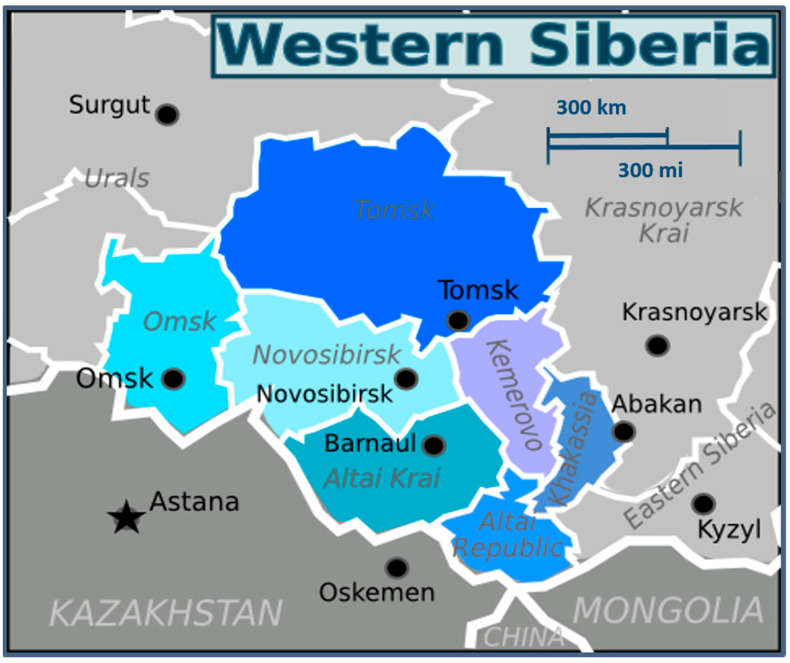
Geographical position of Barnaul and Altai Krai, Western Siberia (modified from https://commons.wikimedia.org/wiki/File:Western_Siberia_WV_map_PNG.png; accessed on 3 July 2023).

**Figure 2 pathogens-12-00914-f002:**
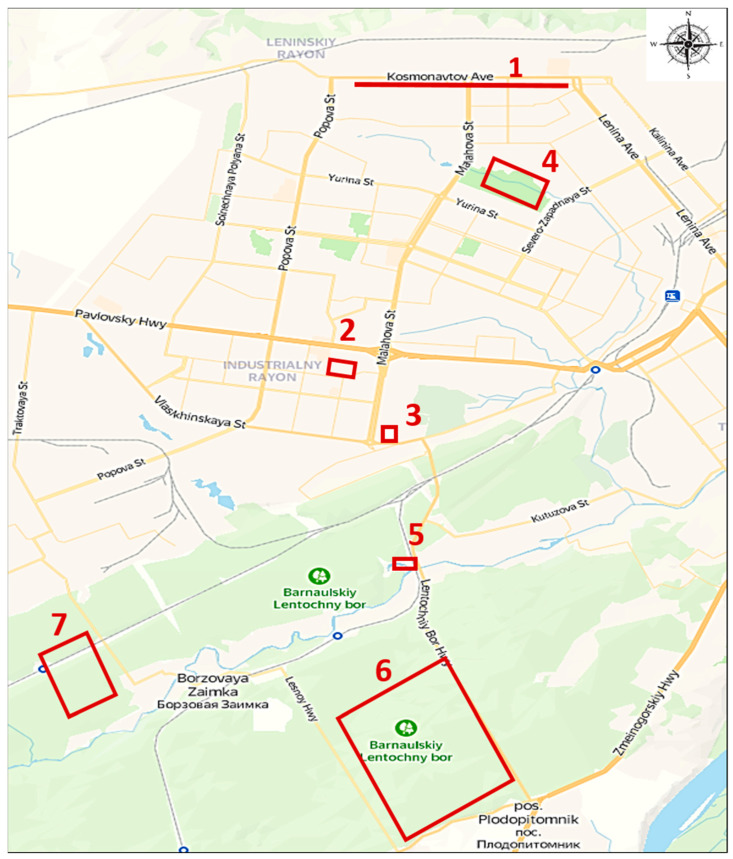
The map of tick collection zones in Barnaul. Zones for tick collection with geographic coordinates: urban ecosystem with grass plantations (1 [53°23′12.0′′ N 83°42′57.7′′ E], 2 [53°20′27.3′′ N 83°41′07.4′′ E]), urban ecosystem near buildings next to coniferous forest (3 [53°19′40.1′′ N 83°41′39.0′′ E]), urban forest park area (4 [53°22′10.0′′ N 83°43′15.8′′ E]), boundary between aquatic and forest ecosystems (5 [53°18′26.7′′ N 83°41′59.1′′ E]), boundary between urban and forest ecosystems (6 [53°16′32.1′′ N 83°42′15.4′′ E], 7 [53°17′16.4′′ N 83°37′34.9′′ E]).

**Figure 3 pathogens-12-00914-f003:**
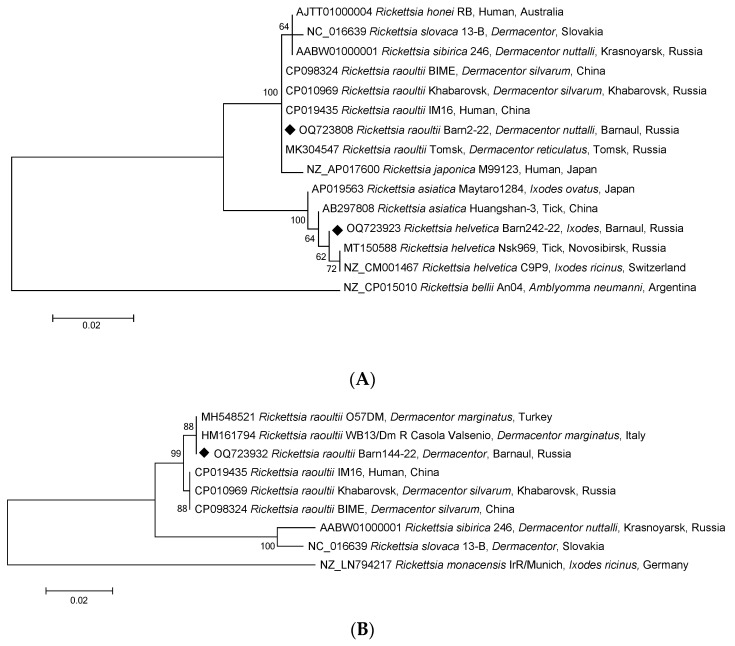
Phylogenetic trees constructed using the maximum likelihood method based on nucleotide sequences of *Rickettsia* spp. from ticks, including ones from this study (Barnaul, black diamonds), and reference sequences of the (**A**) *gltA* (384 bp) and (**B**) *ompA* (532 bp) gene fragments. The *R. bellii* An04 (NZ_CP015010) and *R. monacensis* IrR/Munich (NZ_LN794217) sequences were used as an outgroup. The GenBank accession numbers for reference sequences are shown with the sequence name, tick species, and country. The branch numbers indicate bootstrap support (1000 replicates). The scale bar indicates the phylogenetic distance.

**Figure 4 pathogens-12-00914-f004:**
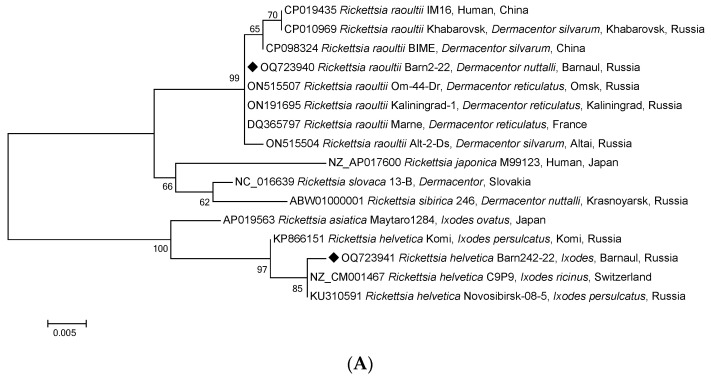
Phylogenetic tree constructed using the maximum likelihood method based on nucleotide sequences of *Rickettsia* spp. from ticks, Barnaul (black diamonds), and reference sequences of the (**A**) *ompB* (427 bp), (**B**) *htrA* (450 bp), and (**C**) 16S rRNA (768 bp) gene fragments. The GenBank accession numbers for reference sequences are shown with the sequence name. The branch numbers indicate bootstrap support (1000 replicates). The scale bar indicates the phylogenetic distance.

**Table 1 pathogens-12-00914-t001:** List of primers used in the study.

Primer Name	Sequence (5′-3′)	Annealing Temperature (°C)	Amplicon Size (bp)	Reference
Rp877p	GGGGACCTGCTCACGGCGG	58	382	[5]
Rp1258n	ATTGCAAAAAGTACCGTGAACA			
Rr.190.70p	ATGGCGAATATTTCTCCAAAA	55	532	[6]
Rr.190.602n	AGTGCAGCATTCGCTCCCCCT			
rompB SFG IF	GTTTAATACGTGCTGCTAACCAA	57	426	[7]
rompB SFG/TG IR	GGTTTTGCCCATATACCGTAAG			
Rr17k.90p	GCTCTTGCAGCTTCTATGTT	55	450	[8]
Rr17k.539n	TCAATTCACAACTTGCCATT			
16S3	GATGGATGAGCCCGCGTCAG	65	772	[3]
16S4	GCATCTCTGCGATCCGCGAC			

**Table 2 pathogens-12-00914-t002:** Prevalence of tick-borne rickettsioses pathogens in ticks, Barnaul, 2022.

Genus	Collecting Zones	Number of Ticks	Number of Ticks Infected by SFGR (%, 95% CI)	
			*R. raoultii*	*R. helvetica*
*Dermacentor*	1	71	37 (52.1, 40.7–63.3)	0
	2	60	33 (55.0, 42.5–66.9)	0
	3	11	11 (100.0, 70.0–100.0)	0
	4	1	0	0
	5	57	42 (73.7, 60.9–83.4)	0
	6	2	2 (100.0, 29.0–100.0)	0
	7	-	-	-
	Subtotal	202	125 (61.9, 55.0–68.3)	0
*Ixodes*	1	-	-	-
	2	-	-	-
	3	-	-	-
	4	11	0	0
	5	5	0	2 (40.0, 11.6–77.1)
	6	39	0	2 (5.1, 0.5–17.8)
	7	42	0	1 (2.4, <0.01–13.4)
	Subtotal	97	0	5 (5.1, 1.9–11.8)
*Haemaphysalis*	7	1	0	0
Total		300	125 (41.7, 36.2–47.3)	5 (1.7, 0.6–4.0)

## Data Availability

The sequences from this study are available in the NCBI GenBank under accession numbers OQ723808-OQ723947 and OQ727556-OQ727559.

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
