# Peer review of "High Prevalence of Rickettsia raoultii Found in Dermacentor Ticks Collected in Barnaul, Altai Krai, Western Siberia"

_pathogens, 2023, doi:10.3390/pathogens12070914_

Round 1

Reviewer 1 Report

Suggested title: A high prevalence of Rickettsia raoultii found in Dermacentor ticks collected in Barnaul, Altai Krai, Western Siberia.

Keywords: must be different than those from the title.

Introduction.

Please, provide split paragraphs into different ideas. Also, provide references for each statement.

Materials and methods.

Please describe the study area, climate, native vegetation structure, geographical coordinates, and a map referencing the city's geographic position in the region and country.

L51-53. These are the results. Provide collecting area description and then collecting methodology. Include conservation and transportation methods. Specify the time between collecting and molecular procedures, and provide collecting time, distances, flag size, and collecting effort for the fieldwork. 

L66-73. Please provide references for supporting molecular methods.

Results.

L89. Please change sampled by collected.

L93. Please, eliminate the sentence. These details are in the M&M section.

L96. What does Ct mean?

L101. "DNA sequences isolated" means that Rickettsia bacteria were isolated using cultures. Is that right? 

Table 2. Please change "sampling" to "collecting." Also, prevalences must have confidence intervals or describe as frequencies.

Suggestion: Include in the discussion section the prevalence and incidence of these rickettsioses in the region, the seasonality of transmission, and the most frequent animal hosts of the studied ticks. 

Can these hosts infect with these Reckettsia? It could be a great point to discuss.

English revision is necessary.

Author Response

Response to Reviewer 1 Comments

On behalf of my co-authors, we thank you very much for giving us an opportunity to revise our manuscript. Your comments are all valuable and very helpful for revising and improving our paper, as well as the important guiding significance to our researches. Below are the point-by-point responses and modifications we made to the manuscript. We have highlighted the new text in the manuscript in yellow.

Point 1. Suggested title: A high prevalence of Rickettsia raoultii found in Dermacentor ticks collected in Barnaul, Altai Krai, Western Siberia.

Response 1. We have modified the title as you suggested. We only omitted “A”.

Point 2. Keywords: must be different than those from the title.

Response 2. Thank you for the point. We changed keywords to the new one: “ticks; spotted fever group Rickettsia; phylogenetic analysis; Russia”.

Point 3. Introduction. Please, provide split paragraphs into different ideas. Also, provide references for each statement.

Response 3. We split the first paragraph and added two new references (#1 and #4).

Point 4. Materials and methods. Please describe the study area, climate, native vegetation structure, geographical coordinates, and a map referencing the city's geographic position in the region and country.

L51-53. These are the results. Provide collecting area description and then collecting methodology. Include conservation and transportation methods. Specify the time between collecting and molecular procedures, and provide collecting time, distances, flag size, and collecting effort for the fieldwork.

Response 4. We have added the requesting information to Section 2.1 and Figure 1.

Point 5. L66-73. Please provide references for supporting molecular methods.

Response 5. DNA extraction and qPCR were performed according to the kit manufacturer's instructions. The conventional PCR is described in more detail in the references (column 5 of Table 1).

Point 6. Results. L89. Please change sampled by collected.

Response 6. Done.

Point 7. L93. Please, eliminate the sentence. These details are in the M&M section.

Response 7. Done.

Point 8. L96. What does Ct mean?

Response 8. In qPCR assay a positive reaction is detected by accumulation of a fluorescent signal. The Ct (cycle threshold) is defined as the number of cycles required for the fluorescence signal to cross the threshold (i.e. exceeds background level) defined in the kit instructions. We have added an explanation to the text “Ct (cycle threshold)”.

Point 9. L101. "DNA sequences isolated" means that Rickettsia bacteria were isolated using cultures. Is that right?

Response 9. No, we did not culture rickettsiae; we have only detected the rickettsial DNA in the tick homogenate by qPCR and conventional PCR. Sentence changed to "DNA sequences found".

Point 10. Table 2. Please change "sampling" to "collecting." Also, prevalences must have confidence intervals or describe as frequencies.

Response 10. Replaced and added 95% confidence intervals. We have also added how we calculate confidence intervals in Materials and Methods.

Point 11. Suggestion: Include in the discussion section the prevalence and incidence of these rickettsioses in the region, the seasonality of transmission, and the most frequent animal hosts of the studied ticks. Can these hosts infect with these Reckettsia? It could be a great point to discuss.

Response 11. We have added the first two new paragraphs to the Discussion.

Reviewer 2 Report

The authors provided background information indicating evidence for significant incidence of tick-borne Rickettsial disease in the Altai Krai region of Siberia and noted that the local prevalence of spotted fever group rickettsial pathogens was not known, resulting in delays in the appropriate diagnosis and onset of treatment for human diseases apparently associated with tick bites. The authors collected 300 ticks, including 202 of genus Dermacentor, 97 Ixodes, and 1 Haemaphysalis. from Barnaul, Altai Krai, Western Siberia and utilized qPCR to screen for the presence of Rickettsial spp. Over 43% of the ticks were classified as positive for Rickettsia spp. Rickettsial marker genes (gltA, ompA, ompB, htrA, and 16S rRNA) were sequenced and compared by BLAST analyses to GenBank for species identification. Results indicated that all of the Rickettsia sequenced for Dermacentor ticks were R. raoultii, while those from Ixodes ticks were identified as R. helvetica. Rickettsia raoulti is thought to be the cause of DEBONEL/TIBOLA/SENLAT syndrome and does not cause the classic rash clinic typical of most tick-borne Rickettsia known to be present in Russia (NATT, Far Eastern tick-borne ricke?siosis, Astrakhan spotted fever). Although R. helvitica is not known to cause human disease, reports from Sweden suggest that it may be the causative agent for fever, without a rash. These Rickettsial species do not usually cause development of a rash that is often used by clinicians to aid in diagnosis of tick-borne Rickettsial disease. Documentation of these Rickettsial species present in Siberian Dermacentor and Ixodes spp. ticks should greatly facilitate appropriate diagnosis and treatment of tick-borne disease in Western Siberia.

The manuscript is well written, with very few errors. I did notice missing spaces in Reference #2, following the year of publication. However, I believe in view of the international readership, please include the sequences of the primers utilized in Secton 2.2 for qPCR screening to identify Rickettsia spp. prior to subsequent sequance analysis (Section 2.3), as the manufacturer and associated information may not be widely available or accessible.

Author Response

Response to Reviewer 2 Comments

We would like to extend our appreciation to you for reviewing our manuscript and for your positive feedback on our work. We sincerely appreciate the time and effort you have dedicated to reviewing our work. Once again, we express our gratitude for your valuable contribution to our manuscript.

Point 1. I did notice missing spaces in Reference #2, following the year of publication.

Response 1. We did not find any missing spaces in the Reference in our version of the manuscript. This is probably one of the formatting issues that could be quickly resolved before publication.

Point 2. However, I believe in view of the international readership, please include the sequences of the primers utilized in Section 2.2 for qPCR screening to identify Rickettsia spp. prior to subsequent sequence analysis (Section 2.3), as the manufacturer and associated information may not be widely available or accessible.

Response 2. The kit used for the qPCR screening is a patented commercial product, and the instructions do not provide any information regarding the sequences of the primers, which is definitely industrial know-how. The only information provided by the manufacturer is the target gene, which is ompB. We have included it in the corresponding sentence.
